# Deep Learning for Laryngopharyngeal Reflux Diagnosis

**Gen Ye** [2,3], **Chen Du** [1], **Tong Lin** [2,3], **Yan Yan** [1,*] and **Jack Jiang** [4]

1   Department of Otolaryngology, Peking University Third Hospital, Beijing 100191, China; dcbriandu@163.com
2   The Key Laboratory of Machine Perception (MOE), School of Electronics Engineering and Computer Science, Peking University, Beijing 100871, China; yegen@pku.edu.cn (G.Y.); lintong@pku.edu.cn (T.L.)
3   Peng Cheng Laboratory, Shenzhen 518052, China
4   Department of Surgery, Division of Otolaryngology–Head and Neck Surgery, University of Wisconsin School of Medicine and Public Health, Madison, WI 53705, USA; jjjiang@wisc.edu
*   Correspondence: yanyan_ent@bjmu.edu.cn; Tel.: +86-13381281669

**Abstract:** (1) Background: Deep learning has become ubiquitous due to its impressive performance in various domains, such as varied as computer vision, natural language and speech processing, and game-playing. In this work, we investigated the performance of recent deep learning approaches on the laryngopharyngeal reflux (LPR) diagnosis task. (2) Methods: Our dataset is composed of 114 subjects with 37 pH-positive cases and 77 control cases. In contrast to prior work based on either reflux finding score (RFS) or pH monitoring, we directly take laryngoscope images as inputs to neural networks, as laryngoscopy is the most common and simple diagnostic method. The diagnosis task is formulated as a binary classification problem. We first tested a powerful backbone network that incorporates residual modules, attention mechanism and data augmentation. Furthermore, recent methods in transfer learning and few-shot learning were investigated. (3) Results: On our dataset, the performance is the best test classification accuracy is 73.4%, while the best AUC value is 76.2%. (4) Conclusions: This study demonstrates that deep learning techniques can be applied to classify LPR images automatically. Although the number of pH-positive images used for training is limited, deep network can still be capable of learning discriminant features with the advantage of technique.

**Keywords:** laryngopharyngeal reflux; laryngoscope; diagnosis; deep learning

## 1. Introduction

Laryngopharyngeal reflux disease (LPR) may involve about 50% of all patients with laryngeal or voice disorders [1]. Misdiagnosis and over-diagnosis of LPR are common phenomena in general otolaryngology clinics. To diagnose LPR both conveniently and effectively is a big challenge for the laryngologists. The Reflux Score Index (RSI) and Reflux Finding Score (RFS) are the subjective diagnostic tools that applied widely today. As for the objective method, the gold standard is the multichannel intraluminal impedance 24 h pH monitoring (MII-24 h pH) currently. However, pH monitoring has several drawbacks, such as aggressive approach, the inconvenience and discomfort for a patient, and the disagreement on the placement of proximal probe. Hence, both scales and MII-24 h pH monitoring are not ideal diagnostic methods, so analyzing laryngoscope images appears to be a promising direction that is convenient, effective and low-cost.

Du et al. proposed to extract color and texture features from seven specific regions in laryngoscope images for distinguishing LPR [2]. However, their approach relies on manually drawn regions to extract features, so large variations might occur during this kind of manual works and this is not an end-to-end approach. Moreover, the single hidden-layer network used in their approach may not capture hierarchical structures in laryngoscope images. Litjens et al. showed that deep learning with multilayer neural networks had permeated the entire field of medical image analysis [3]. Therefore, in this work, we focus on leveraging deep learning methods in an end-to-end fashion for an LPR diagnostic system [4]. However, it is well-known that deep learning requires massive data to be

effective, which may pose a great challenge for this medical image application due to the time-consuming and costly endeavor of collecting data and making annotations for the dataset. This work explored how recent deep-learning approaches perform on this small-sample problem.

## 2. Materials and Methods

### 2.1. Data Collection

This experiment was conducted with the approval from the ethics committee of the hospital. 152 subjects from the hospital were studied prospectively from March 2015 to April 2019. All subjects were evaluated subjectively by a reflux symptom index (RSI) and a laryngoscopy. The subjects holding no LPR-related symptoms served as our control group. With their consents, the subjects with LPR-related symptoms also underwent 24 h pH probe monitoring.

The pH monitoring results indicate that 37 subjects were confirmed with LPR, and 38 subjects were suspected with LPR but negative. The control group was composed of 77 healthy subjects. All subjects in this study were of Chinese descent and divided into three groups as follows.

The pH-positive group consisted of 37 subjects such that proximal acid episodes $\geq 3$, or proximal acid exposure time >1%, or impedance detected proximal acid exposure $\geq 4$. There was no specific RSI cutoff for this group.

The pH-negative group consisted of 38 subjects such that proximal acid episodes < 3, or proximal acid exposure time $\leq 1\%$, or impedance detected proximal acid exposure < 4. There was no specific RSI cutoff for this group.

The control group consisted of 77 subjects who were administered laryngoscopy for routine test before thyroid surgery and without LPR-related symptoms and who had an RSI $\leq 13$.

Laryngeal images were obtained during a laryngoscopy through a flexible high-resolution laryngoscope (model EVNE; XION; Berlin, Germany). The images were taken near the middle of the larynx, and the white balance function was employed to exclude other potential factors that may interfere with the true image colors. All the images were encoded with the JPEG format. In the study, we only consider the two-class problem of pH-positive cases versus control cases, which is the most distinguishable scenario with total number of examples $n = 114$.

### 2.2. Backbone Network

A residual network (ResNet) [5], a famous convolutional neural network architecture, was used as our backbone network. A ResNet stacks several same building blocks (see Figure 1) as the feature extractor and employs a fully connected layer as its final classifier. A ResNet can be constructed with different depth depending on the number of building blocks. In our study, each residual block conducts the following computation:

$$y = f(x, \{W_1, W_2\}) + x$$

where $x$ and $y$ indicate the input and output of the block. The function $f(x, \{W_1, W_2\}) := W_2\sigma(W_1x)$, where the activation function $\sigma$ is a rectified linear unit (ReLU) [6]. The batch normalization (BN) [7] was adopted between the convolution layer and the ReLUs and the weights of BN were omitted in the former equation for simplifying notations.

Most of our experiments used ResNet18 (with 18 layers) as the backbone network. Each image was resized and cropped into the size of $224 \times 224$ as inputs. The network finally outputted a single value $\in [0, 1]$, treated as the confidence in that the input was pH positive. Instead of using a predefined fixed threshold, a varying threshold that determines the prediction of each image was adopted so that the prediction accuracy could be calculated. By manipulating the threshold value, a receiver operating characteristic (ROC) curve was gotten to measure the sensitivity of our model.

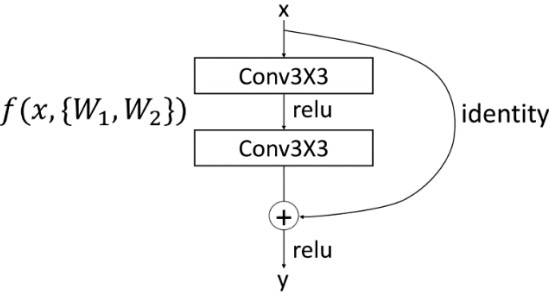

**Figure 1.** The building block of a residual network.

### 2.3. Attention Modules

We attempted to enhance the backbone with two recently proposed attention approaches: Squeeze-and-Excitation module (SE) [8] and convolution block attention module (CBAM) [9], which enabled the network to focus on particular channels or spatial areas of interests. A Squeeze-and-Excitation (SE) block estimated the importance of each channel by learning from data. Each SE block (Figure 2a) conducted the following computation:

$$y = F_{scale}(MLP(AvgPool(x)), x)$$

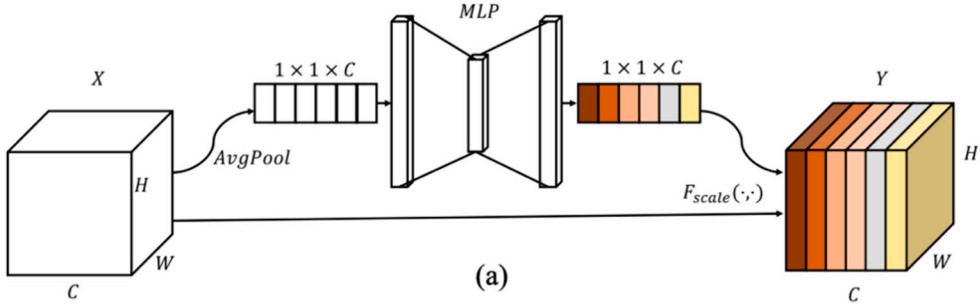

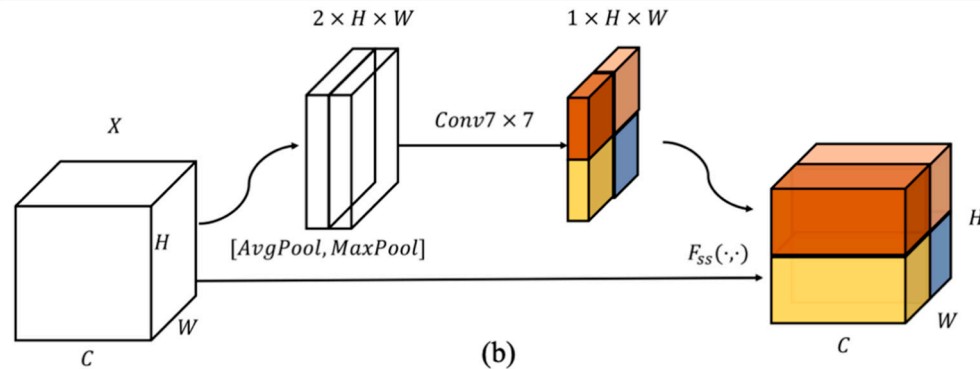

**Figure 2.** (**a**). A Squeeze-and-Excitation (SE) block. (**b**). A spatial sub-module of the convolution block attention module (CBAM).

Specifically, average pooling $AvgPool()$ was adopted for obtaining channel-wise statistics. Then the statistics of all channels were passed to a two-layer fully connected network MLP and transformed into an output vector indicating the importance of each channel. Finally, $F_{scale}$ referred to a channel-wise multiplication with the obtained importance vector.

A CBAM block estimated the importance of each channel and each spatial area by inferring a 1D channel attention map $AC \in R^{C \times 1 \times 1}$ and a 2D spatial attention map $AS \in$

$R^{1 \times H \times W}$ (see Figure 2b). Here H, W and C denoted the numbers of height, width and channels of the feature maps, respectively. A CBAM module conducted the following computations:

$$A_C(x) = \sigma(MLP(AvgPool(x)) + MLP(MaxPool(x))),$$

$$A_S(x) = Conv_{7 \times 7}([AvgPool(x), MaxPool(x)]),$$

$$y' = F_{CS}(A_C(x), x),$$

$$y = F_{SS}(A_S(y'), y').$$

where $\sigma$ denoted the sigmoid function and $Conv_{7 \times 7}$ referred to a convolution layer with kernel size of $7 \times 7$. $F_{CS}$ and $F_{SS}$ meant the functions performing element-wise scaling using two type of attention values. It was worth noting that the average pooling and the maximum pooling function in two attention maps were also performed across different dimensions.

*2.4. Dealing with Data Deficiency*

During the training phase, our datasets were augmented by sequentially applying random rotation, color jittering, random flip and random crop. During the testing phase, our datasets were augmented by the same type of transformations but in a deterministic manner, obtaining 55 images for each original image. Then all the augmented images were fed to the classification network and 55 predictions were obtained for each original image. The final decision was based on majority voting of these multiple predictions. Empirically we found that such majority voting scheme could enhance the classification performance.

Besides data augmentation and majority voting, transfer learning [10,11] and metric-based few-shot learning [12] were also adopted to our task for overcoming data scarcity. Transfer learning intends to use the knowledge or data from another relative domain to ease the learning of the target domain. Few-shot learning methods focus particularly on the learning problem with very limited data.

In the study, two transfer learning strategies were adopted. One strategy was simply using a pretrained network as a feature extractor, while another attempted to fine-tune the pretrained network on the target domain further. In transfer learning experiments, our models were pretrained using the ImageNet dataset from ILSVRC-2012 [13].

For few-shot learning, a feature extractor trained from the miniImageNet dataset [14] was used and applied on our images with the same procedure proposed by Snell et al. [12]. For generating a prediction, the input image and a support set from training images were mapped to a common feature space, and then the distance from the input feature to prototype representations of each class was computed for classification. This few-shot learning method could be summarized as follows:

$$c_k = \frac{1}{N_K} \sum_{x \in S_K} f(x),$$

$$p(y = k|x) = \frac{exp(-dist(f(x), c_k))}{\sum_{k'} exp(-dist(f(x), c_{k'}))}.$$

where the prototype representation $c_k$ of the class k is the mean of features of support set samples from class k, while f denotes the feature map. The distance function dist $(\cdot, \cdot)$ denoted squared Euclidean distance. The probability of the input image belonging to class k was computed by a softmax function based on the distance vector.

**3. Results**

*3.1. Backbone Network with Attention Modules*

In this backbone experiment, nine models were trained: ResNet18, ResNet50, ResNet101 and their enhancement versions with SE or CBAM attention modules. Multiple residual blocks were stacked to construct our backbone network. We attempted to find a proper backbone network in two directions: modifying the depth of network and enhancing the

network with newly invented attention modules. ResNet18 (with 18 layers), ResNet50 (with 50 layers) and ResNet101 (with 101 layers) were trained for finding the proper network depth. The SE module and CBAM module were integrated to the backbone network for testing the usefulness of attention modules. Architectures of some models are shown in Table 1.

**Table 1.** The architectures of main backbone models.

| Feature Size | ResNet18 | ResNet18_SE | ResNet18 _CBAM | ResNet50 | ResNet101 |
|---|---|---|---|---|---|
| 112 × 112 | | | 7 × 7, 64, stride 2 | | |
| 56 × 56 | | | max pooling, 3 ×3, stride 2 | | |
| | $\begin{bmatrix} 3 \times 3,\ 64 \\ 3 \times 3,\ 64 \end{bmatrix} \times 2$ | $\begin{bmatrix} 3 \times 3,\ 64 \\ 3 \times 3,\ 64 \\ f_C,\ [4,\ 64] \end{bmatrix} \times 2$ | $\begin{bmatrix} 3 \times 3,\ 64 \\ 3 \times 3,\ 64 \\ f_C,\ [4,\ 64] \\ 7 \times 7 \end{bmatrix} \times 2$ | $\begin{bmatrix} 1 \times 1,\ 64 \\ 3 \times 3,\ 64 \\ 1 \times 1,\ 256 \end{bmatrix} \times 3$ | |
| 28 × 28 | $\begin{bmatrix} 3 \times 3,\ 128 \\ 3 \times 3,\ 128 \end{bmatrix} \times 2$ | $\begin{bmatrix} 3 \times 3,\ 128 \\ 3 \times 3,\ 128 \\ f_C,\ [8,\ 128] \end{bmatrix} \times 2$ | $\begin{bmatrix} 3 \times 3,\ 128 \\ 3 \times 3,\ 128 \\ f_C,\ [8,\ 128] \\ 7 \times 7 \end{bmatrix} \times 2$ | $\begin{bmatrix} 1 \times 1,\ 128 \\ 3 \times 3,\ 128 \\ 1 \times 1,\ 512 \end{bmatrix} \times 4$ | $\begin{bmatrix} 1 \times 1,\ 128 \\ 3 \times 3,\ 128 \\ 1 \times 1,\ 512 \end{bmatrix} \times 4$ |
| 14 × 14 | $\begin{bmatrix} 3 \times 3,\ 256 \\ 3 \times 3,\ 256 \end{bmatrix} \times 2$ | $\begin{bmatrix} 3 \times 3,\ 256 \\ 3 \times 3,\ 256 \\ f_C,\ [16,\ 256] \end{bmatrix} \times 2$ | $\begin{bmatrix} 3 \times 3,\ 256 \\ 3 \times 3,\ 256 \\ f_C,\ [16,\ 256] \\ 7 \times 7 \end{bmatrix} \times 2$ | $\begin{bmatrix} 1 \times 1,\ 256 \\ 3 \times 3,\ 256 \\ 1 \times 1,\ 1024 \end{bmatrix} \times 6$ | $\begin{bmatrix} 1 \times 1,\ 256 \\ 3 \times 3,\ 256 \\ 1 \times 1,\ 1024 \end{bmatrix} \times 23$ |
| 7 × 7 | $\begin{bmatrix} 3 \times 3,\ 512 \\ 3 \times 3,\ 512 \end{bmatrix} \times 2$ | $\begin{bmatrix} 3 \times 3,\ 512 \\ 3 \times 3,\ 512 \\ f_C,\ [32,\ 512] \end{bmatrix} \times 2$ | $\begin{bmatrix} 3 \times 3,\ 512 \\ 3 \times 3,\ 512 \\ f_C,\ [32,\ 512] \\ 7 \times 7 \end{bmatrix} \times 2$ | $\begin{bmatrix} 1 \times 1,\ 512 \\ 3 \times 3,\ 512 \\ 1 \times 1,\ 2048 \end{bmatrix} \times 3$ | $\begin{bmatrix} 1 \times 1,\ 512 \\ 3 \times 3,\ 512 \\ 1 \times 1,\ 2048 \end{bmatrix} \times 3$ |
| 1 × 1 | | | average pooling, 1D fc, softmax | | |

The brackets indicate the content of each basic building block, while the numbers outside the brackets mean the duplication times of stacked blocks. The number of full-connection (fc) layer indicates the output dimension of channel attention module, and the $7 \times 7$ layer indicates the kernel size of a spatial attention module.

Five-fold cross-validation was adopted to report the test performance of these models. That was, each time, all images were randomly partitioned into five chunks with equal sizes, where four chunks (80% images) were taken as training data and the remaining chunk (20% images) was left as test data. We ran three times of five-fold cross validation to get the averaged classification accuracy and AUC value of each model. All of our models were trained via stochastic gradient descent (SGD). The adaptive learning rate method Adam [15] was used with an initial learning rate of $10^{-3}$. Each model was trained for 500 episodes. The batch size was chosen to be 8.

The results are shown in Table 2. We can see that, without any attention module, ResNet50 performed better than ResNet18 and ResNet101. When attention module was allowed to incorporate into the backbone network, ResNet50 with the CBAM module achieved the best classification accuracy, at 73.4%, while ResNet18 with the CBAM module offered the best AUC value, at 73.9%.

**Table 2.** Test performance of the backbone models (mean $\pm$ SD).

| Model | Accuracy | AUC Value |
|---|---|---|
| ResNet18 | $0.688 \pm 0.089$ | $0.704 \pm 0.095$ |
| ResNet18_SE | $0.697 \pm 0.086$ | $0.715 \pm 0.123$ |
| ResNet18_CBAM | $\mathbf{0.729 \pm 0.050}$ | $\mathbf{0.739 \pm 0.068}$ |
| ResNet50 | $0.707 \pm 0.126$ | $\mathbf{0.719 \pm 0.117}$ |
| ResNet50_SE | $0.659 \pm 0.088$ | $0.680 \pm 0.116$ |
| ResNet50_CBAM | $\mathbf{0.734 \pm 0.101}$ | $0.719 \pm 0.132$ |
| ResNet101 | $0.672 \pm 0.108$ | $0.678 \pm 0.115$ |
| ResNet101_SE | $0.656 \pm 0.074$ | $0.682 \pm 0.103$ |
| ResNet101_CBAM | $\mathbf{0.679 \pm 0.110}$ | $\mathbf{0.691 \pm 0.096}$ |

Figure 3 depicted the accuracy curves during training procedures. The left subfigure compared the performance of different depth. We could see that with more epochs the training accuracy of each model keeps improving, while the test accuracy starts to fluctuate from 150 epochs. Moreover, ResNet50 and RetNet18 performed similarly on test data at most epochs, but they offered higher accuracy than ResNet101. The right subfigure shows the training procedures of ResNet18 with and without attention modules. Clearly the ResNet18 with CBAM attention modules achieved the best test accuracy when compared to the ResNet18 backbone and the ResNet18 with SE attention modules.

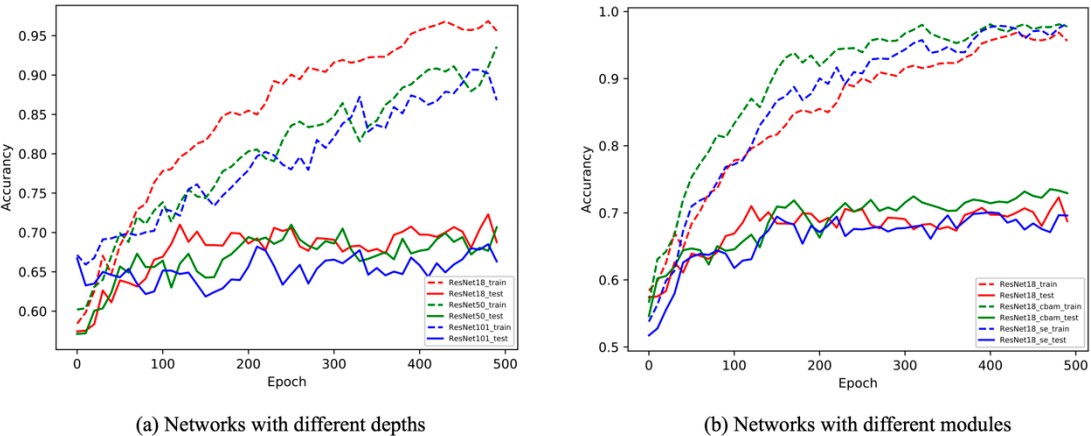

(a) Networks with different depths          (b) Networks with different modules

**Figure 3.** Training and testing accuracies of networks. The curves were obtained by averaging the accuracies of three times of five-fold cross validations. (**a**) Networks with different depths. (**b**) Networks with different modules.

According to the above observations, the ResNet18 was chosen as our backbone network in the following, since it was lightweight and performed decently compared with other deep networks.

### 3.2. Transfer Learning

The transfer learning techniques was leveraged to enhance the performance of the backbone network ResNet18 and ResNet18 CBAM. The backbone network was pretrained on the image classification task with the large ImageNet dataset. Then we fine-tuned it on our laryngeal image dataset in two distinct ways: (1) The first residual block was frozen as the feature extractor and learned the weights of following layers with relatively small learning rate $10^{-4}$(denoted as ResNet18 + freeze and ResNet18 CBAM + freeze). (2) All the weights of the entire network were modified with learning rate $10^{-4}$(denoted as ResNet18 + finetune and ResNet18 CBAM + finetune).

In this setting, six models were trained: ResNet18, ResNet18 CBAM and their freeze or finetune versions. Except for the learning rate, all hyperparameters were set in the same way as the previous backbone network experiments. The results of accuracy and AUC

value are shown in Table 3. The ResNet18 + freeze model yielded a higher accuracy and AUC value than its original ResNet18. However, both freeze and finetune variants could not improve the performance of ResNet18 CBAM, as seen with the training procedure shown in Figure 4.

**Table 3.** Test performance of transfer learning models.

| Model | Accuracy | AUC Value |
|---|---|---|
| ResNet18 | $0.688 \pm 0.089$ | $0.704 \pm 0.095$ |
| ResNet18 + freeze | $\mathbf{0.702 \pm 0.112}$ | $\mathbf{0.762 \pm 0.106}$ |
| ResNet18 + finetune | $0.676 \pm 0.078$ | $0.694 \pm 0.078$ |
| ResNet18_CBAM | $\mathbf{0.729 \pm 0.050}$ | $\mathbf{0.739 \pm 0.068}$ |
| ResNet18_CBAM + freeze | $0.593 \pm 0.158$ | $0.659 \pm 0.122$ |
| ResNet18_CBAM + finetune | $0.626 \pm 0.096$ | $0.724 \pm 0.103$ |

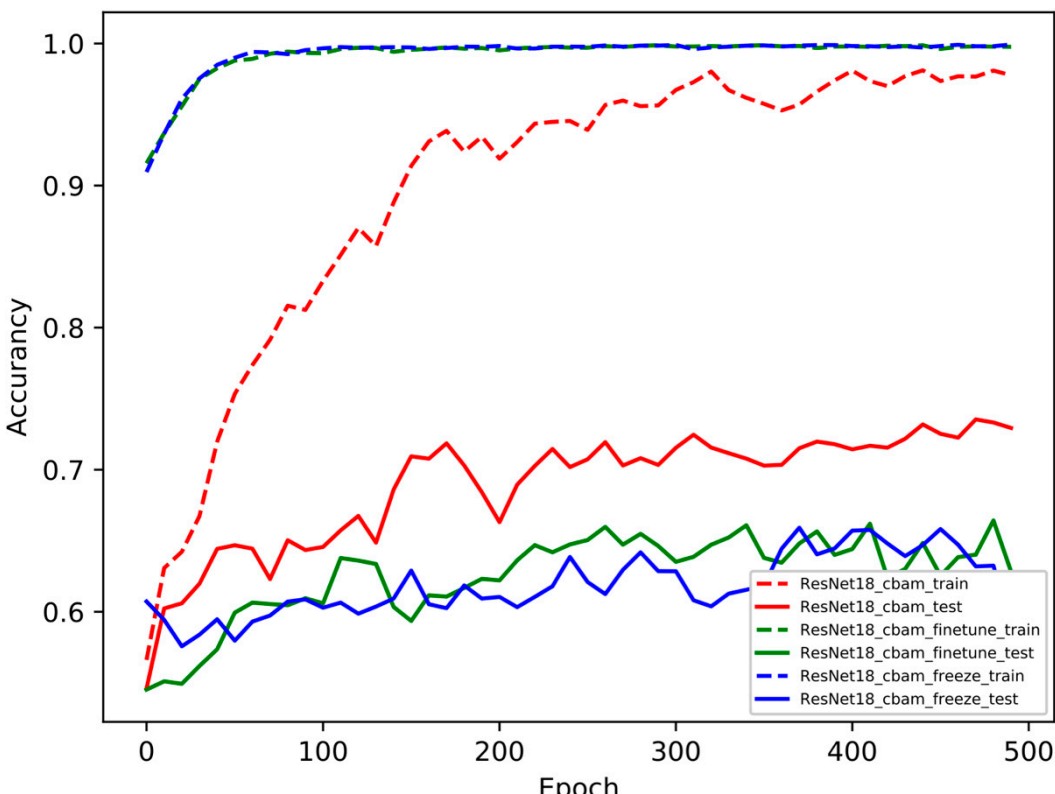

**Figure 4.** Training curves of transfer learning models. The curves were obtained by the same way as Figure 3.

### 3.3. Few-Shot Learning

The few-shot learning was also employed to cope with the small sample problem in the laryngoscopic image classification task. We argue that, due to the limited size of training data, only a feature extractor (a shallow network) could be trained rather than a complete classifier (a deep network). In our experiment, the shallow network was composed of four convolution layers and a fully connected output layer, as in a previous work [12].

Two models were trained: ProtoNet and ProtoNet + finetune. The weights of the former model were trained by using the miniImageNet dataset, while the weights of the latter were fine-tuned by using LPR images additionally. The results are shown in Table 4. Although fine-tuning enhanced the performance, the second model was still outperformed by previous backbone models. This experiment shows that prototype-based few-shot learning cannot succeed in improving LPR diagnosis performance.

**Table 4.** Test performance of few shots learning models.

| Model | Accuracy | AUC Value |
| --- | --- | --- |
| ProtoNet | 0.578 | 0.610 |
| ProtoNet + finetune | **0.632** | **0.623** |

## 4. Discussion

Deep learning techniques have been applied to several mainstream medical images, such as CT, MRI and ultrasound. However, only a few methods are devoted to endoscope images or laryngoscope images particularly. The hypothesis was that raw laryngoscope images can be used to diagnose LPR directly and conveniently, which has been verified by previous works [2,16]. Following those works, MII-24 h pH monitoring was adopted as the gold standard to diagnose LPR, which was more accurate and objective than the RFS method.

The results of Tables 2 and 3 indicate that deep neural networks are promising to capture discriminant representations for the LPR task. Although an overall classification accuracy of 73.4% may be inadequate for real settings of clinical diagnosis, a relatively small image dataset would be responsible, which included 114 subjects with only 37 pH-positive cases. As the database is enlarged in the future, deep learning techniques will achieve a better performance.

Another contribution of this work is that the extensive experiments were conducted to test the performance of some prominent deep neural networks on this particular task. A bunch of baselines were set up for future study, to build LPR diagnostic system with deep learning techniques. Unfortunately, based on the data of this research, the popular techniques in transfer learning and few-shot learning were found to be unable to enhance the performance further. On the other hand, the deep learning technique is a kind of quick diagnostic tool for LPR, which can be applied before and after therapy to evaluate the effect more conveniently than MII-24 h pH monitoring.

Du et al. proposed an LPR diagnostic method where predefined texture and color features were extracted on seven manually drawn regions [2]. In our work, discriminant features are learned automatically in an end-to-end fashion, which allows a clinician to sidestep the tedious work and eliminate the subjective bias due to manual drawing jobs. With the technology development of deep learning, the features of laryngoscope image could be recognized from every angle of view, not just with only two features, as color and texture. Furthermore, as the datasets were augmented by sequentially applying random rotation, color jittering, random flip and random crop during the training and testing phase, the effect of interference factors, such as the distance, angle of the camera and white balance of laryngoscope images, would be eliminated. The future work will involve testing more powerful network architectures, particularly those new models in attention mechanism, transfer learning and few-shot learning.

When the data are enough for the deep learning system, the performance of the technique will be better. The collection of images from multi-center is a good choice.

## 5. Conclusions

This study demonstrates that deep learning techniques can be applied to classify LPR images automatically. Although only with limited pH-positive images for training, deep networks can still be capable of learning discriminant features to generalize on unseen images. One advantage of deep learning techniques over traditional techniques is that, as we collect more images to enlarge our dataset in the future, the performance might be improved with the same training procedure with little extra efforts.

**Author Contributions:** Conceptualization, Y.Y., T.L. and J.J.; methodology, T.L.; software, G.Y.; formal analysis, G.Y.; investigation, C.D.; resources, C.D.; writing—original draft preparation, G.Y.; writing—review and editing, C.D.; supervision, Y.Y. and T.L.; funding acquisition, Y.Y. All authors have read and agreed to the published version of the manuscript.

**Funding:** This research was funded by "Inter-disciplinary Funding of Medicine of Peking University" (No. BMU2020MI007), NSFC Tianyuan Fund for Mathematics (No. 12026606) and National Key R&D Program of China (No. 2018AAA0100300).

**Institutional Review Board Statement:** The study was conducted according to the guidelines of the Declaration of Helsinki and approved by the ethics committee of University of Wisconsin-Madison and Peking University Third Hospital.

**Informed Consent Statement:** Informed consent was obtained from all subjects involved in the study.

**Data Availability Statement:** All data generated or analyzed during this study are included in this published article.

**Conflicts of Interest:** The authors declare no conflict of interest. The funders had no role in the design of the study; in the collection, analyses or interpretation of data; in the writing of the manuscript; or in the decision to publish the results.

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
