# Peer review of "Deep Learning for Laryngopharyngeal Reflux Diagnosis"

_applsci, doi:10.3390/app11114753_

Round 1
Reviewer 1 Report
The paper entitled “Deep Learning for Laryngopharyngeal Reflux Diagnosis” proposes this innovative method to evaluate an ENT common disease.
The topic is very interesting and it could be relevant in clinical practice. However, I have some doubts:
Abstract:
Line 14: it is described a prior work based on RFS or pH monitoring? It is referred to literature or to a particular work? in this case, could you better define it?
Introduction:
The starting sentence “Nearly 50% of all patients with laryngeal or voice disorders may suffer from laryngopharyngeal reflux disease (LPR)” should be converted in: “Laryngopharyngeal reflux disease may involve about 50% of all patients with laryngeal or voice disorders”.
Materials and Methods:
Line 55: patients were evaluated by RSI and laryngoscopy, therefore was RFS calculated? If it was not performed, clarify your reason, especially because it remains the most used and inexpensive method in LPR diagnosis.
Ex: The RFS does not take into consideration many LPR findings, including vocal fold erythema, leukoplakia, posterior pharyngeal wall inflammation, anterior pillars inflammation, and coated tongue (please cite DOI: 10.1177/0194599819827488).
Line68: How have you excluded LPR in the control group? Have you performed pH-metry?
Results:
Data of deep learning performance are well expressed, but none is reported about the laryngeal evidence. It would be better to show RSI values for each group and the hue values of each region measured individually.
Discussion:
Have you evaluated the possibility to use deep learning to compare larynx images captured by NBI?
Ex: In recent years, the international literature indicated that NBI laryngoscopy is useful for diagnosing not only laryngeal neoplastic disease but also inflammatory disease and that NBI could improve the identification of LPR signs (please cite doi: 10.23812/20-314-L).

Author Response
Abstract:
- Line 14: Was the described prior work based on RFS or pH monitoring? It is referred to in the literature or to a particular work? in this case, could you better define it?
The prior work that we refer to is past research. In other words, LPRD is diagnosed based on RSI, RFS, or 24-hour pH monitoring.
Introduction:
- The sentence starting with “Nearly 50% of all patients with laryngeal or voice disorders may suffer from laryngopharyngeal reflux disease (LPR)” should be converted in: “Laryngopharyngeal reflux disease may involve about 50% of all patients with laryngeal or voice disorders”.
We agree with this correction and have made the change in the manuscript.
Materials and Methods:
- Line 55: patients were evaluated by RSI and laryngoscopy, therefore was RFS calculated? If it was not performed, clarify your reason, especially because it remains the most used and inexpensive method in LPR diagnosis.
Ex: The RFS does not take into consideration many LPR findings, including vocal fold erythema, leukoplakia, posterior pharyngeal wall inflammation, anterior pillars inflammation, and coated tongue (please cite DOI: 10.1177/0194599819827488).
Indeed, the RFS is the most used and inexpensive method in LPR diagnosis. However, there are only 8 signs that are evaluated, which cannot describe all of the details in a laryngoscopic image. Our work intends to analyze every detail of the image through deep learning to diagnose and evaluate LPR more precisely. Thus, RFS was not shown here.
- Line 68: How have you excluded LPR in the control group? Have you performed pH-metry?
The control group subjects were patients who underwent laryngoscopy for other reasons, such as the routine exam before thyroid surgery, and had no symptoms related to LPR. RSI score confirmed that they were not suspected LPRD patients. So, pH-metry was not performed.
Results:
- Data of deep learning performance are well expressed, but none is reported about the laryngeal evidence. It would be better to show RSI values for each group and the hue values of each region measured individually.
Of course, there is much to be improved in our research. In this study, laryngoscopic images were not partitioned for deep learning. In the future, the drawing of each region is going to be done automatically, which would eliminate the subjective bias.
Discussion:
- Have you evaluated the possibility to use deep learning to compare larynx images captured by NBI?
Ex: In recent years, the international literature indicated that NBI laryngoscopy is useful for diagnosing not only laryngeal neoplastic disease but also inflammatory disease and that NBI could improve the identification of LPR signs (please cite doi: 10.23812/20-314-L).
That is a good suggestion! In future, we can use deep learning to analyze NBI images to see if there were meaningful findings. However, this aspect is not relevant to the work presented in this manuscript, so it was not included.

Reviewer 2 Report
Dear authors!
The topic is quite interesting and the paper itself is written quite ok.
I have a few concersns / questions outlined below:
1.) The group size of patients with diagnosed or suspected LPR is a bit too low. Especially if you want to prove the value of "deep learning".
Maybe you could collect more data.
2.) Why did you not use the Belafsky Reflux Finding Score (RFS) for assessment during laryngoscopy!? That would have created more standardised data. Results of laryngoscopy are a bit dependent on examiner experience without a score.
3.) You mentioned, that you have performed 24h-pH-monitoring in the group of 37 patients, you have suspected to suffer from LPR.
24h-pH-monitoring is still the goldstandard, but alone not 100% reliable to diagnose LPR. Did you use the DeMeester score to differentiate between patients with LPR and healthy controls?!
Author Response
The topic is quite interesting and the paper itself is written quite ok.
I have a few concerns / questions outlined below:
- The group size of patients with diagnosed or suspected LPR is a bit too low. Especially if you want to prove the value of "deep learning". Maybe you could collect more data.
We agree that the group size is a bit low. While Peking University Third Hospital is large, there is not much data on pH monitoring of positive patients. We suspect that LPRD patients are afraid to carry out pH monitoring.
We made efforts to resolve this problem. First, we augmented the data sets by sequentially applying random rotation, color jittering, random flips, and random crops during the training and testing phase. Second, we are still collecting the LPR patients by pH-monitoring and laryngoscope images.
- Why did you not use the Belafsky Reflux Finding Score (RFS) for assessment during laryngoscopy!? That would have created more standardised data. Results of laryngoscopy are a bit dependent on examiner experience without a score.
While RFS is the most used and inexpensive method in LPR diagnosis, only 8 signs are evaluated, which cannot describe all the details of a laryngoscopic image. Our work intends to analyze every detail of the laryngoscope image through deep learning to diagnose and evaluate LPR more precisely. Thus. RFS was not shown here.
- You mentioned that you have performed 24h-pH-monitoring in the group of 37 patients, you have suspected to suffer from LPR.
24h-pH-monitoring is still the gold standard, but alone not 100% reliable to diagnose LPR. Did you use the DeMeester score to differentiate between patients with LPR and healthy controls?!
The DeMeester score is primarily used to identify patients with GERD. Thus, we didn’t use DeMeester score in this study.

Round 2
Reviewer 1 Report
Authors clarified the required points. Therefore, the papers deserves to be accepted.
Reviewer 2 Report
Dear authors!
I recommend publication.